# Size Reduction to Enhance Crystal-to-Liquid Phase Transition Induced by *E*-to-*Z* Photoisomerization Based on Molecular Crystals of Phenylbutadiene Ester

**DOI:** 10.3390/ma17153664

**Published:** 2024-07-24

**Authors:** Yu-Hao Li, Min Cui, Yi Gong, Tian-Yi Xu, Fei Tong

**Affiliations:** Key Laboratory for Advanced Materials and Joint International Research Laboratory of Precision Chemistry and Molecular Engineering, Feringa Nobel Prize Scientist Joint Research Center, Frontiers Science Center for Materiobiology and Dynamic Chemistry, School of Chemistry and Molecular Engineering East China University of Science and Technology, 130 Meilong Road, Shanghai 200237, China; yuhao_li@mail.ecust.edu.cn (Y.-H.L.); cuimin2021@163.com (M.C.); 22011017@mail.ecust.edu.cn (Y.G.); xutianyi1108@163.com (T.-Y.X.)

**Keywords:** crystal engineering, molecular crystals, photoinduced melting, nanomaterials

## Abstract

Harnessing the photoinduced phase transitions in organic crystals, especially the changes in shape and structure across various dimensions, offers a fascinating avenue for exact spatiotemporal control, which is crucial for developing future smart devices. In our study, we report a new photoactive molecular crystal made from (*E*)-2-(3-phenyl-allylidene)malonate ((*E*)-PADM). When exposed to ultraviolet (UV) light at 365 nm, this compound experiences an *E*-to-*Z* photoisomerization in liquid solution and a crystal-to-liquid phase transition in solid crystals. Remarkably, nanoscopic crystalline rods boost their melting rate and degree compared to bulk crystals, indicating that miniaturization enhances the photoinduced melting effect. Our results demonstrate a simple approach to rapidly drive molecular crystals into liquids via photochemical reactions and phase transitions.

## 1. Introduction

Photoinduced phase transition materials refer to those that can undergo a phase change or transformation under light illumination through chemical reactions. These materials typically can transform from one phase (such as solid) to another phase (such as liquid, gaseous, or different crystalline forms) under suitable lighting conditions [1,2,3,4]. Light, as a clean energy source and a non-contact external stimulus, can be regulated to achieve precise spatiotemporal resolution and manipulation of responsive materials [5,6,7,8,9,10]. Current research on photoinduced phase transition materials mainly focuses on organic molecular crystals [11,12,13,14,15], liquid crystal elastomers [16,17], and polymeric materials [18,19,20]. Liquid crystal elastomers often exhibit order only in one or two dimensions, which means that the degree of phase transition in photo-responsive materials based on liquid crystal elastomers is inferior to that of three-dimensionally ordered organic molecular crystal materials [16,17]. However, the preparation of liquid crystal elastomers involves cumbersome steps such as polymerization, which requires the process of forming a polymer from monomer molecules, and crosslinking, which requires chemically bonding polymer chains together, requiring strict control over molecular weight and imposing high demands on molecular design and synthesis.

On the other hand, photoresponsive polymer materials are obtained by doping photosensitive molecules with photoresponsive properties into polymeric networks [18,19,20]. The doping amount of photosensitive molecules is generally low to ensure stability, resulting in insufficient phase transition and deformation extents in these materials. Compared with the two materials mentioned above, organic molecular crystal materials can be used directly after synthesis without additional polymerization and doping processes. Molecular crystals possess an ordered anisotropic packing structure and high Young’s modulus, which are capable of producing significant morphological and shape deformations during the phase transition process. Xu et al. reported a molecular crystal (DNAM) that can undergo efficient photoisomerization (*E*-to-*Z*) under visible light illumination, accompanied by rapid melting phase transitions (crystalline to liquid state) [11]. The surface areas of the micrometer-thin sheets shrink on average by approximately 97%, with a maximum reduction of up to 98.5% after light illumination. Thus, organic molecular crystals are photoresponsive phase transition materials with good application potential.

Currently, research on the photoinduced phase transitions of organic molecular crystals is limited. It mainly focuses on azobenzene and its derivatives [12,13,14,15] due to their reversible photoisomerization behavior accompanied by significant phase transition. Taking advantage of the photoinduced phase transition (crystal to amorphous), Hao et al. observed independent photomechanical actions in azobenzene derivatives under different temperatures when illuminating the crystals [13]. However, despite possessing good photoinduced phase transition performance and significant morphological changes, the development of organic molecular crystal materials still faces challenges. The difficulty arises from the fact that bulk molecular crystals might undergo disintegration or shattering due to the intense internal strains produced during a photochemical reaction [21,22,23,24,25]. Moreover, inherent factors such as molecular and crystal structure, size, shape, and morphology also profoundly influence the properties of crystals [26,27,28,29,30,31]. Norikane et al. [14] confirmed that the photoinduced crystal-to-liquid transformation speed of azobenzene derivatives in the film was more than 30 times faster than that of powder crystals, which indicated that reducing the size could enhance the photoinduced phase transition performance of organic molecular crystals. The solvent annealing method in anodic aluminum oxide (AAO) template used by Bardeen et al. [32,33,34,35] can reduce the molecular crystal size to the nanometer level. If this method is introduced into photoinduced phase transition organic molecular crystals, their performance could be significantly improved.

Herein, we designed and synthesized a novel butadiene derivative molecule named (*E*)-2-(3-phenyl-allylidene)malonate ((*E*)-PADM). When exposed to ultraviolet (UV) light (365 nm), bulk polycrystals of (*E*)-PADM with dimensions larger than 1 mm in length and width undergo a slow crystal-to-liquid phase transition, partially converting into liquid after approximately 2.5 h of light exposure. The phase transition rate significantly increases when the bulk crystals are ground into smaller powder crystals, producing more liquid in just a few minutes. However, under the same light exposure conditions, nanoscopic (*E*)-PADM wires completely melt into liquid within one second, approximately 9000 times faster than their bulk counterparts. Our results demonstrate that the size reduction dramatically accelerates the speed of photoinduced phase transitions in crystals. Our results offer a straightforward approach to achieving dramatic photoresponsive behaviors in organic molecular crystals at the microscopic level, in contrast to those at the macroscopic scale. The rapid phase transition feature in nanoscopic crystals can potentially be applied in future miniature intelligent switches and energy transducers.

## 2. Materials and Methods

### 2.1. Materials

All starting materials were purchased from J&K Scientific Company (Shanghai, China) and used without further purification. The cetyltrimethylammonium bromide (CTAB), *n*-decyl glucoside (DG), 3-(*N,N*-dimethyldodecylammonio)propane sulfonate (SB-12), and sodium dodecyl sulfate (SDS) were purchased from Sigma-Aldrich (Shanghai, China) (≥99%). The anodic aluminum oxide (AAO) template was purchased from TOP Membranes Technology Co. Ltd. (Shenzhen, China). The organic solvents used were of analytical reagent grade (A.R.), and Milli-Q water (18 MΩ/cm) was used throughout the experiments. More detailed information on material synthesis can be found in the Supporting Materials.

### 2.2. Measurements and Characterization

The ^1^H NMR and ^13^C NMR data were both collected from a superconducting Fourier NMR spectrometer Bruker Advance-III HD 600 MHz (Bruker, Ettlingen, Germany), using deuterated dimethyl sulfoxide-d6 (DMSO-*d*_6_) and deuterated acetone-d6 as solvents at 298 K. High-resolution time-of-flight mass spectrometry (HR-ESI) was performed using an LCT Premier XE mass spectrometer from the Waters Company of the Milford, MA, USA. The ultraviolet–visible absorption spectra were recorded with a Shimadzu UV-3600 Plus spectrophotometer (Shimadzu, Kyoto, Japan). The optical microscopy measurements were conducted using a Leica DM750 microscope equipped with a QTF500 digital camera (QTF, Yixing, China). The microscopy fluorescence measurements were performed using a TL-3201LED fluorescence microscope (equipped with a 20 MP USB 3.0 digital camera) (Dilun Optical, Shanghai, China). All PXRD data were collected on a Rigaku D/Max 2550 VB/PC X-ray powder diffractometer (Rigaku, Tokyo, Japan, CuKα radiation, step size: 0.02°, 5–75°) at room temperature. Single-crystal XRD data were collected on a Double target micro focal spot single-crystal X-ray diffraction system (D8 Venture (Mo), Bruker, Germany) at 213 K. The thermogravimetric analysis (TGA) was performed on a Shimadzu TGA-50/50H thermal analyzer (Shimadzu, Japan). The differential scanning calorimetry (DSC) analysis was performed on a Shimadzu DSC-60 Plus analyzer (Shimadzu, Japan).

### 2.3. Synthesis of (E)-2-(3-phenylallylidene)malonic Acid ((E)-PAPA)

A total of 2.080 g of malonic acid (0.020 mol) was dissolved in 10 mL of pyridine, and the reaction setup was placed in an ice bath. Separately, 1.320 g of cinnamaldehyde (0.010 mol) was dissolved in 10 mL of pyridine. After the setup had been completely cooled, the cinnamaldehyde solution was added, followed by gradual mixing in 1.0 mL of 4-methylpiperidine (0.006 mol). Under argon atmosphere (after three cycles of argon replacement), the ice bath was removed and the reaction mixture was left at room temperature for 12 h. After this period, 280 mL hydrochloric acid solution (3.4 M) was added into the reaction mixture to form a yellow precipitate. Stirring continued for 30 min, and then, the mixture was extracted with EtOAc until clear. The combined organic layer was washed with brine three times, dried over anhydrous MgSO_4_, and concentrated in vacuo to obtain a yellow solid. Finally, the crude product was dissolved in 20 mL of acetic acid and hexane was slowly added for recrystallization. After drying, the yellow solid powder of (*E*)-PAPA (1.71 g, yield 78.44%) was obtained. (Appendix A). ^1^H NMR (600 MHz, DMSO-*d*_6_) δ (ppm): 7.54 (d, *J* = 7.0 Hz, 2H), 7.43–7.36 (m, 4H), 7.26 (d, *J* = 15.5 Hz, 1H), 7.17 (dd, *J* = 15.5, 11.4 Hz, 1H) (Appendix A). ^13^C NMR (151 MHz, DMSO-*d*_6_) δ (ppm): 166.61, 165.90, 143.61, 142.51, 135.51, 129.75, 129.06, 127.58, 126.94, 123.25 (Appendix A). HR-MS (ESI): calculated for [C_12_H_9_O_4_]^−^, [M]^−^ = 217.0506, found = 217.0502 (Appendix A).

### 2.4. Synthesis of Dimethyl (E)-2-(3-phenylallylidene)malonate ((E)-PADM)

A total of 1.090 g (0.005 mol) of (*E*)-PADA was dissolved in 25 mL of anhydrous N,N-dimethylformamide (DMF), and the same molar amount of potassium carbonate (6.900 g) was added to the Schlenk flask. Under an argon atmosphere (after three cycles of replacement), the reaction flask was sealed with aluminum foil, and 2.5 mL of iodomethane (5.700 g, 0.040 mol) was added through a syringe in the dark, followed by refluxing for 12 h. After the reaction was complete, the reaction mixture was washed with brine three times, dried over anhydrous MgSO_4_, and concentrated in vacuo. Then, the concentrated mixture was purified by column chromatography (PE: EtOAc, 9:1, *v*/*v*), and a pale-yellow oily liquid was collected. The crude product was dissolved in dichloromethane, recrystallized slowly with methanol, and dried to obtain white granules (*E*)-PADM (0.79 g, yield 64.23%).(Appendix A). ^1^H NMR (600 MHz, Acetone-*d*_6_) δ (ppm): 7.61 (d, *J* = 6.8 Hz, 2H), 7.54 (dd, *J* = 8.9, 1.8 Hz, 1H), 7.44–7.38 (m, 3H), 7.31–7.25 (m, 2H), 3.86 (s, 3H), 3.78 (s, 3H) (Appendix A). ^13^C NMR (151 MHz, Acetone-*d*_6_) δ (ppm): 155.44, 154.68, 135.63, 135.02, 125.58, 120.39, 119.33, 118.26, 114.45, 113.06, 42.75, 42.67 (Appendix A). HR-MS (EI): calculated for [C_14_H_14_O_4_], [M] = 246.0892, found = 246.0894 (Appendix A).

### 2.5. Preparation of (E)-PADM Single-Crystals for Structure Determination

A total of 26.9 mg of (*E*)-PADM solid powders was dissolved into 200 μL of dry DMF to form a 0.5 M homogenous solution. The solution was placed in an open sample bottle at room temperature. After the solvent was fully evaporated (~72 h), (*E*)-PADM single crystals for structure determination were harvested.

### 2.6. Preparation of (E)-PADM Nanowires

A total of 13 mg of (*E*)-PADM powder solid was dissolved into 60 μL DMF. The solution was carefully dropped onto an AAO template, allowing it to spread completely on the surface of the template. It was then sealed with a glass cover, and the solvent was left to anneal at 55 °C for 4 days. After annealing, the surface of the AAO template was gently polished with sandpaper until smooth. Then, the template was dissolved in a 0.1%/20% (wt%) SDS/H_3_PO_4_ solution for one day to obtain dispersed nanowires (Appendix A).

## 3. Results

We first explored the photochemical reaction of (*E*)-PADM molecules in solution (tetrahydrofuran (THF) as the solvent) via UV–Vis absorption spectroscopy measurements (Figure 1a). The spectrum of the (*E*)-PADM solution exhibited a broad absorbance band, ranging from approximately 275 nm to 375 nm, centering at around 323 nm. This absorbance band corresponds to the S_0_→S_1_ transition of the (*E*)-PADM molecule. When the solution was exposed to UV light (365 nm), the absorbance of (*E*)-PADM molecules at 323 nm gradually decreased, with a blue shift to 316 nm, dropping by about 30% after 120 s. In the meantime, an isosbestic point appeared at 290 nm, suggesting the characteristics of *E*-*Z* isomerization. After 120 s of light irradiation, the absorbance track did not change, indicating that the photochemical reaction in the solution reached a photostationary state. The ^1^H NMR spectra of the solution under different illumination durations were also investigated in Figure 1b. After irradiation at 365 nm, a characteristic peak of the *Z* isomer appeared and increased at 7.1 ppm. In contrast, the height of the *E* isomer decreased at 7.6 ppm, which indicates the continuous progress of *E*-*Z* isomerization. By calculating the hydrogen integration ratios at these two sites, it was found that after 10 min of irradiation, the conversion rate of *E*-*Z* isomerization was approximately 44.8%. After 30 min of irradiation, the photoproduct was about 63.9%. The experimental UV–Vis spectral and chemical shifts were also consistent with the computational results (Appendix A).

The photochemical properties of (*E*)-PADM solids were also investigated. Since the (*E*)-PADM powders yielded a sticky liquid after a photochemical reaction, it was impossible to measure their UV–Vis absorption spectra directly. To address this, we dissolved the photoproduct obtained from different illumination times into THF to prepare dilute solutions of the same concentration for solution measurements. The UV–Vis absorption spectrum of the solution based on the initial (*E*)-PADM solids without illumination was consistent with that in the solution state (Appendix A). Moreover, when the (*E*)-PADM powders were exposed to 365 nm light, the absorbance at 323 nm decreased, showing an about 5 nm blue shift. Compared with the spectra in the solution, the spectra based on solid samples exhibited a more prominent decline and blue shift, with an isosbestic point appearing at around 273 nm. A characteristic peak of the *Z* isomer still appeared at 7.1 ppm according to the ^1^H NMR measurements. When the illumination time reached 120 min, the ratio of (*E*)-PADM to (*Z*)-PADM was 50.5%:49.5%. However, the signal peak from 7.2 ppm to 7.5 ppm bulged after illumination, which might belong to the generated oligomers of the (*E*)- and (*Z*)-PADM molecules as multiple peaks were observed. We investigated the HR-MS of photoproducts to verify our hypothesis. As shown in Appendix A, not only the peaks of the monomer but also the peaks of its dimers, trimers, and other oligomers were observed in the HR-MS. Notably, it is worth mentioning that the UV–Vis absorption spectra and ^1^H NMR of the solution and solid after irradiation with visible light or heating did not change, indicating that the photoisomerization of (*E*)-PADM at 365 nm is irreversible.

The higher molecular structural symmetry of the *E* isomer leads to a more ordered crystal arrangement than that of the *Z* isomer, resulting in a higher melting point for the *E* isomer. The *Z* isomer had a lower melting point, even below room temperature, which leads to the liquid-like photoinduced melting phenomenon after isomerization. The thermogravimetric analysis (TGA) and differential scanning calorimetry (DSC) measurements were conducted on (*E*)-PADM powder crystals to determine the decomposition temperature and melting point (Figure 2a,b). Since the chemical properties of the *E* and Z forms are similar and difficult to separate, the viscous liquid of the Z form is challenging to weigh for testing. We also performed the same analysis on a photoproduct mixture after a photostationary state under 365 nm irradiation (Figure 2c,d). The TGA curve of (*E*)-PADM illustrated that (*E*)-PADM had good thermal stability with a high decomposition temperature (220 °C), and the melting point of (*E*)-PADM could be determined as 68 °C according to the DSC curve. As for the photoproduct, no crystallization peak had been found in the DSC curve under −70 °C. The difference in melting points between the raw material and the product was over 100 °C, which could explain the photoinduced melting behavior of (*E*)-PADM.

To clarify the reason for oligomer formation, we first studied the single-crystal structure of (*E*)-PADM. As shown in Figure 3, the single crystal of the (*E*)-PADM belongs to the *Pccn* space group, and (*E*)-PADM molecules adopt a herringbone packing structure. A solid-state [2 + 2] cycloaddition between adjacent C=C double bonds requires the distance to be less than 4.2 Å, according to the Schmidt topochemistry principle [36]. However, the distance of adjacent double bonds in (*E*)-PADM is about 7.063 Å, which indicates that the [2 + 2] cycloaddition of (*E*)-PADM monomeric molecules would not be possible. Compared to the (*E*)-PADM, the *Z*-isomer is more distorted, which would be even pronounced when in a liquid state. We inferred that the other photoproducts except the *Z*-isomer, including dimer, trimer, and other oligomers found in the photoproducts, according to the HR-MS measurements, could be formed by introducing the isomerized product (*Z*)-PADM or chain polymerization of double bonds during the crystal-to-liquid phase transition process. Unfortunately, we could not obtain the single crystals of the (*Z*)-PADM due to the low melting point of (*Z*)-PADM (<−70 °C).

The optical and polarized microscope images of *(E)*-PADM bulk crystals (length and width >1 mm) and powder crystals (length and width ~100 μm) before and after light exposure are shown in Figure 4a–d and Appendix A. It could be observed that the crystals exhibited strong birefringence before light exposure, indicating good crystallinity. After being exposed to 365 nm light for 2.5 h, the rigid edges of the bulk crystals became slightly soft, showing partial crystal-to-liquid melting phase transition as the birefringence still existed in the crystals after exposure (Figure 4a,b and Appendix A). When the crystal size shrunk to crystalline powders, the light irradiation duration for a significant melting became 10 min. After light irradiation, the photoproduct still exhibited birefringence under the cross-polarized microscope, indicating incomplete photoinduced melting in the powder crystals (Figure 4c,d and Appendix A).

We also prepared nanoscopic crystalline (*E*)-PADM wires by adopting the solvent annealing method in anodic alumina oxide (AAO) templates, resulting in uniformly distributed crystals (Figure 4e and Appendix A). The average length of the (*E*)-PADM nanowires was 40 μm, and the average diameter of the nanowires was 200 nm. Upon exposure to 365 nm light, the nanowires instantaneously melted and collapsed into a tiny droplet, with the entire process completed within one second. Unlike the previous two types of crystals, the birefringence phenomenon disappeared in the nanowires after light exposure, indicating that the nanowires had completely transformed from a crystalline state into an amorphous state after illumination (Figure 4e,f and Appendix A). The experimental results demonstrated that the reduction in crystal size significantly shortened the time required for photoinduced melting and caused the crystal structure to collapse, increasing the degree of phase transition. The excitation photons likely penetrated the smaller nanowires deeper, producing more photoproducts that induce a crystal-to-liquid transition.

We also assigned the crystal planes of the (*E*)-PADM nanowire crystal structure. Powder X-ray diffraction (PXRD) measurements were conducted on nanowires filtered on an AAO template and on those cultivated but not dissolved afterward. We compared the results with the theoretically calculated Miller indices obtained from the single-crystal structure. The resulting PXRD patterns are shown in Figure 5 and Appendix A. The (020) crystal facet was arranged vertically, indicating the molecular arrangement in the short-axis direction. The (102) and (204) crystal facets represented the arrangement of the crystal along the long axis. They were crystal planes with different inter-planar distances in the same direction, both perpendicular to (020), which was consistent with the objective fact that the long and short axes of the crystal are perpendicular to each other. In the horizontal direction, there were still (020) and (004) crystal facets, which were due to a small portion of the template not being entirely dissolved when the nanowires were filtered on the AAO template, as well as some nanowires stacking on top of each other at a certain angle to the horizontal direction. As a result, the (*E*)-PADM nanowires were probably single crystals that could undergo fast photochemical reaction and instant melting phase transition once upon light irradiation.

## 4. Conclusions

We have successfully designed and synthesized a new photoactive phenyl-based molecule, with promising potential for practical applications like solar-energy storage, sensors, smart switches, and soft actuators, especially at the microscopic level. This molecule exhibits *E*-*Z* photoisomerization characteristics in both the solution and solid states, with a conversion rate of 63.9% in the solution state and 49.5% in the solid state. Under irradiation with 365 nm light, (*E*)-PADM powder crystals undergo a significant photoinduced melting phase transition, transforming from a highly ordered crystalline state to an amorphous non-crystalline state. We have obtained nanowire crystals using solvent annealing to optimize this photoinduced melting process. The rate of photoinduced melting has been enhanced by more than 9000 times due to the reduction in crystal size. Our results highlight the critical impact of scale effects on photoinduced melting responses and provide new ideas for designing and optimizing photoresponsive materials at the microscopic level, inspiring further research and development in this field.

## Figures and Tables

**Figure 1 materials-17-03664-f001:**
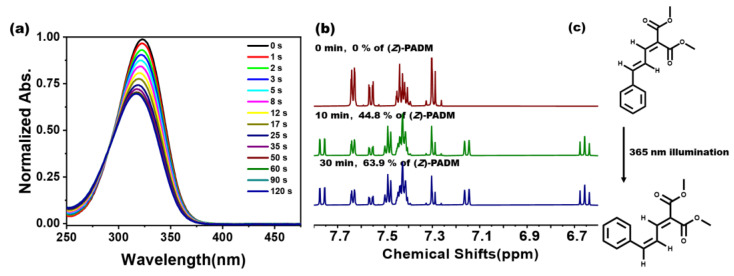
(**a**) The UV–Vis absorption spectrum of (*E*)-PADM in THF solution (1 × 10^−5^ M) after exposure to 365 nm light (light intensity 2.0 mW/cm^2^) for different durations; (**b**) ^1^H NMR spectra of (*E*)-PADM solution in acetone-*d*_6_ (0.041 M) after different irradiation times. (**c**) Scheme of photoisomerization of (*E*)-PADM in THF solution.

**Figure 2 materials-17-03664-f002:**
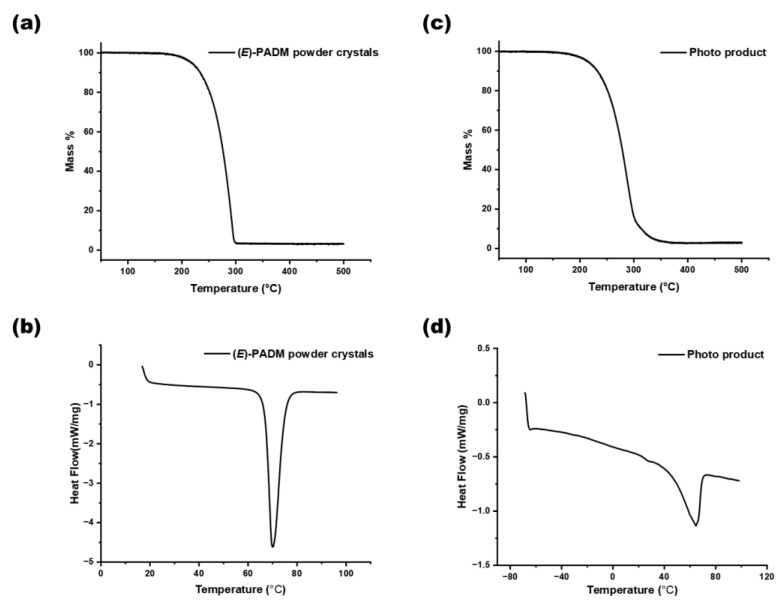
(**a**,**b**) The TG and DSC curve of (*E*)-PADM powder crystals. (**c**,**d**) The TG and DSC curve of the photo product after 365 nm, 20 W illumination for 2 h.

**Figure 3 materials-17-03664-f003:**
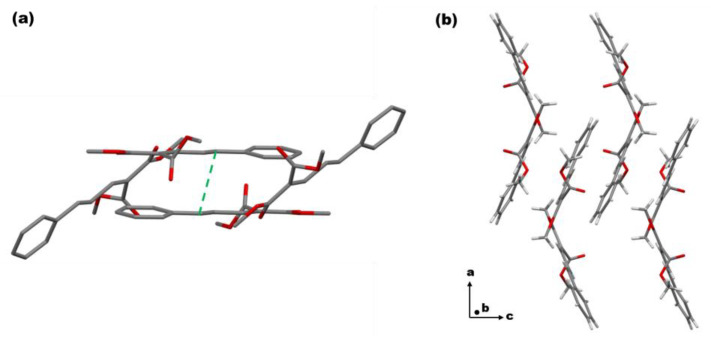
(**a**) Crystal packing structure between paired monomeric molecules of (*E*)-PADM. Hydrogen atoms were omitted for the sake of clarity. (**b**) Single-crystal structure of (*E*)-PADM observed along the crystallographic *b* axis.

**Figure 4 materials-17-03664-f004:**
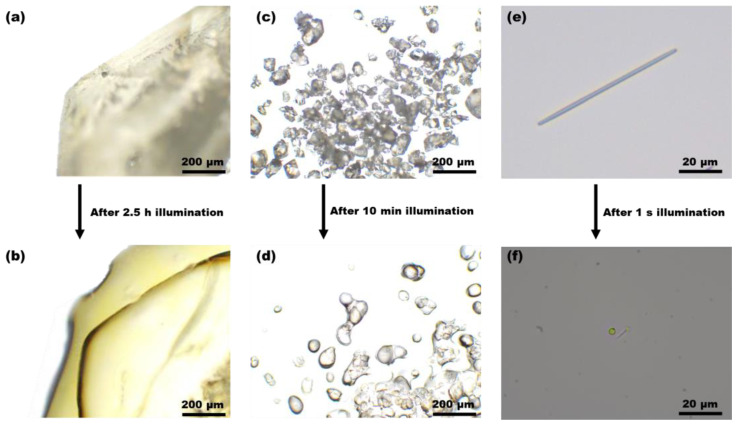
Optical microscope images of (*E*)-PADM crystals: (**a**) a bulk crystal before illumination and (**b**) the bulk crystal after 2.5 h of illumination, with a scale bar of 200 μm; (**c**) powder crystals before illumination and (**d**) powder crystals after 10 min of illumination, with a scale bar of 200 μm; and (**e**) a nanowire crystal before illumination and (**f**) the nanowire crystal after 1 s of illumination, with a scale bar of 20 μm.

**Figure 5 materials-17-03664-f005:**
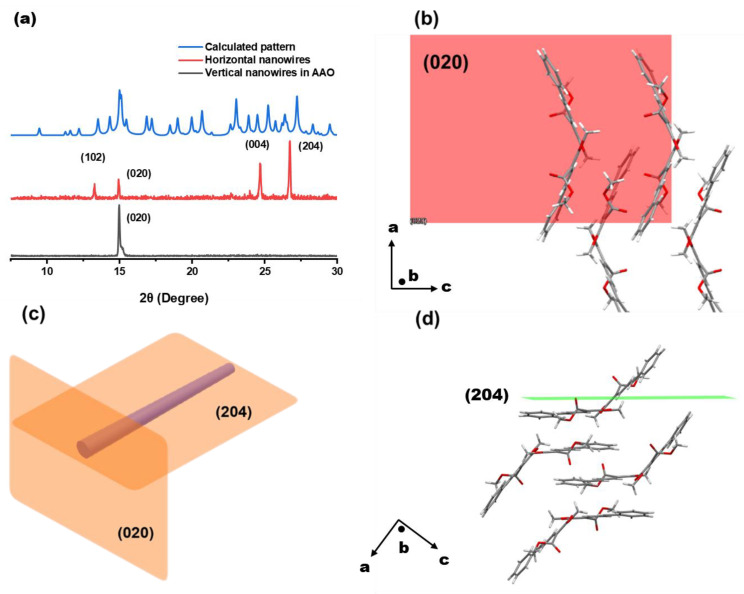
(**a**) The (*E*)-PADM monomer theoretical value calculated from single-crystal data (blue line), the PXRD pattern obtained from the (*E*)-PADM nanowire crystals laid horizontally (red line), and the PXRD pattern obtained from the (*E*)-PADM nanowire crystals laid vertically inside the AAO template (black line). (**b**) The molecular orientation of (*E*)-PADM observed from the top of the nanowire (i.e., under the (020) crystal face). (**c**) Schematic diagram of the (020) and (204) crystal facets of the nanowire crystal. (**d**) The molecular orientation of (*E*)-PADM observed from the side of the nanowire (i.e., under the (204) crystal face).

## Data Availability

The data are contained within the article or Appendix A.

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
