# Peer review of "Size Reduction to Enhance Crystal-to-Liquid Phase Transition Induced by E-to-Z Photoisomerization Based on Molecular Crystals of Phenylbutadiene Ester"

_materials, 2024, doi:10.3390/ma17153664_

Round 1
Reviewer 1 Report
Comments and Suggestions for Authors
In this manuscript, the authors explore the E-Z photoisomerization of the molecule (E)-2-(3-phenyl-allylidene)malonate. While this molecule has been previously synthesized through different routes (the molecule or crystal structure itself is not novel), a study of this particular conversion has not been carried out to my knowledge. My comments and corrections are listed below.
Line 44-45 - remove "organic used directly after synthesis without additional"
Line 47 - remove "which are"
Line 58 - don't capitalize the C in Crystal
Line 63 - change crystal to crystals
Lines (74-83), (236-246), and (247-258) - The statements in these paragraphs are vague and need to be updated. Approximate crystal sizes should be included in these paragraphs to help improve the readers understanding the differences in sizes that are being discussed during the photoreactions.
Line 97 - This instrument is typically list as a Bruker Advance-III HD 600 MHz (Bruker, Germany)
Line 105 - This instrument is typically listed as a Rigaku D/Max 2550 VB/PC (Rigaku, Japan). For pXRD experiments, it is also common to list the step size in degrees and the total scan window (start and stop) 2theta values for the experiments as well.
Line 109 - The X-ray source for the single crystal experiment is noted as Ga, but in the CIF it is recorded as Mo. Please fix this inconsistency. Also in the CIF, the experiment data was collected at 213K, but in the manuscript it is stated that the data was collected at 200K. Please fix this inconsistency as well.
Lines (114-130), (152-158), and 214 - These paragraphs and sentences have multiple grammatical errors that need to be corrected.
In Figure 2, why does the TGA of (E)-PADM go to 0 and the Photo product does not. Something is wrong with these images and they should be corrected.
In Table S1 in the SI, the space group Pccn should be italicized.
Figure 4. Why were the images for the "nanowire crystals" changed in the SI when the images for the bulk crystalline material and "powder crystal" are the same.
For the PXRD experiments, the experimental data and the predicted data from the single crystal experiment don't line up very well. Since the single crystal was collected at 213K, this is not surprising, but an appropriate shift in the predicted data should be made. I would also list the HKL planes on the predicted data as well. For the image in the manuscript, I would zoom in on the area of interest, instead of using a zoomed out graph. In the SI I would include the full diffractograms for the "nanowire" crystals as well as the other bulk crystalline phases to confirm purity.
In the manuscript, you claim that the conversion is based on the size of the crystals. However, you make no direct measurements to confirm this statement. All we see is that a crystal with a very small volume converts/melts faster than crystals with a much larger volume. A better comparison might be to mechanically grind the bulk crystals so that blocky crystals with the similar volume (~5-10 um on a side) could be tested as well.
Comments on the Quality of English LanguageIssues with grammar were addressed in the previous section, however another read through before publication would be advised.
Author Response
Reviewer 1:
In this manuscript, the authors explore the E-Z photoisomerization of the molecule (E)-2-(3-phenyl-allylidene)malonate. While this molecule has been previously synthesized through different routes (the molecule or crystal structure itself is not novel), a study of this particular conversion has not been carried out to my knowledge. My comments and corrections are listed below.
i)Line 44-45 - remove "organic used directly after synthesis without additional"
Line 47 - remove "which are"
Line 58 - don't capitalize the C in Crystal
Line 63 - change crystal to crystals
Response: we sincerely thank for the reviewer for the detailed suggestion on grammar and spelling issues. Therefore, we have carefully checked and revised these errors in our revised manuscript.
- ii) Lines (74-83), (236-246), and (247-258) - The statements in these paragraphs are vague and need to be updated. Approximate crystal sizes should be included in these paragraphs to help improve the readers understanding the differences in sizes that are being discussed during the photoreactions.
Response: we agree with the reviewer that crystal sizes should be in these paragraphs. Thus, we added the measured size in those paragraphs to improve better understanding in the differences in sizes, which is as follows:
“The optical and polarized microscope images of (E)-PADM bulk crystals (length and width >1 mm) and powder crystals (length and width ~100 μm) before and after light exposure are shown in Figure 4a-4d and Figure S13.”
“The average length of (E)-PADM nanowires was 40 μm, and the average diameter of nanowires was 200 nm.”
iii) Line 105 - This instrument is typically listed as a Rigaku D/Max 2550 VB/PC (Rigaku, Japan). For pXRD experiments, it is also common to list the step size in degrees and the total scan window (start and stop) 2theta values for the experiments as well.
Response: In the manuscript, we have revised the description of instruments according to the suggestion of the reviewer.
- iv) The X-ray source for the single crystal experiment is noted as Ga, but in the CIF it is recorded as Mo. Please fix this inconsistency. Also in the CIF, the experiment data was collected at 213K, but in the manuscript it is stated that the data was collected at 200K. Please fix this inconsistency as well.
Response: We are very grateful for the reviewer's suggestions. We have modified the relevant content in the manuscript to make it consistent with the CIF file.
- v) Lines (114-130), (152-158), and 214 - These paragraphs and sentences have multiple grammatical errors that need to be corrected.
Response: We are very grateful for the reviewer's careful examination. We have carefully checked and corrected the relevant grammatical errors.
- vi) In Figure 2, why does the TGA of (E)-PADM go to 0 and the Photo product does not. Something is wrong with these images and they should be corrected.
Response: We agree with the reviewer that the TGA image should be corrected. We re-did the TGA measurements and updated the new data in Figure 2a-2b.
vii) In Table S1 in the SI, the space group Pccn should be italicized.
Response: We have italicized the space group in SI.
viii) For the PXRD experiments, the experimental data and the predicted data from the single crystal experiment don't line up very well. Since the single crystal was collected at 213K, this is not surprising, but an appropriate shift in the predicted data should be made. I would also list the HKL planes on the predicted data as well. For the image in the manuscript, I would zoom in on the area of interest, instead of using a zoomed out graph. In the SI I would include the full diffractograms for the "nanowire" crystals as well as the other bulk crystalline phases to confirm purity.
Response: Thank you for the reviewer's very pertinent suggestions. We have zoomed in the area of interest in Figure 5a and added more detailed data to Figure S14 in the SI. We hope our modifications help readers better understand the relevant data.
viii) In the manuscript, you claim that the conversion is based on the size of the crystals. However, you make no direct measurements to confirm this statement. All we see is that a crystal with a very small volume converts/melts faster than crystals with a much larger volume. A better comparison might be to mechanically grind the bulk crystals so that blocky crystals with the similar volume (~5-10 um on a side) could be tested as well.
Response: We agree with the reviewer's comments and have made the relevant attempts. The powder crystals mentioned in our manuscript were already ground, and if they were ground again, we did not obtain the ideally sized crystals similar to nanowires, but rather some amorphous powder. Therefore, we chose to use the powder crystals as a reference.

Reviewer 2 Report
Comments and Suggestions for Authors
The manuscript entitled "Size Reduction to Enhance Crystal-to-Liquid Phase Transition Induced by E-to-Z Photoisomerization Based on Molecular Crystals of Phenylbutadiene Ester" submitted by Fei Tong and co-workers for consideration for publication in the MDPI journal Materials, presents the investigation of a new photoactive phenyl-based molecule (E)-PADM. When exposed to ultraviolet (UV) light at 365nm, this compound experiences an E-to-Z photoisomerisation in liquid solution and a crystal-to-liquid phase transition in solid crystals. The presented subject is interesting, but the presented data and its presentation need major revision before possible acceptance or transfer. The present subject would be more suitable for MDPI Crystals.
1. 13C NMR spectra should be checked, it is not possible to conduct the measurement with 600 MHz for 13C
2. TGA measurement must be repeated; the mass is increasing at the beginning of the measurements, so it should be constant.
3. What is the reason that there is no char residue at the end of TGA measurements? How is it possible?
4. The conclusion part must be more developed.
5. The novelty and practical applications of the presented data must be highlighted.
Author Response
Reviewer 2:
The manuscript entitled "Size Reduction to Enhance Crystal-to-Liquid Phase Transition Induced by E-to-Z Photoisomerization Based on Molecular Crystals of Phenylbutadiene Ester" submitted by Fei Tong and co-workers for consideration for publication in the MDPI journal Materials, presents the investigation of a new photoactive phenyl-based molecule (E)-PADM. When exposed to ultraviolet (UV) light at 365nm, this compound experiences an E-to-Z photoisomerisation in liquid solution and a crystal-to-liquid phase transition in solid crystals. The presented subject is interesting, but the presented data and its presentation need major revision before possible acceptance or transfer. The present subject would be more suitable for MDPI Crystals.
We thank the very positive comments from the reviewer. We discussed each comment and question individually, along with our corresponding responses.
1.13C NMR spectra should be checked, it is not possible to conduct the measurement with 600 MHz for 13C
Response: We thank the careful review and updated the relevant data. The 13C NMR were measured with 151 MHz.
2) TGA measurement must be repeated; the mass is increasing at the beginning of the measurements, so it should be constant.
Response: We agree with the reviewer that the TGA image should be corrected. We re did the TGA measurements and updated the new data in Figure 2a-2c.
3) What is the reason that there is no char residue at the end of TGA measurements? How is it possible?
Response: we agree with the reviewer that the TGA data has to be revised. We have repeated the TGA measurements and updated the new data in Figure 2a-2c.
4) The conclusion part must be more developed.
Response: Thanks for the reviewer’s suggestion. We have updated the conclusion accordingly, which is as follows:
“We have successfully designed and synthesized a new photoactive phenyl-based molecule, (E)-PADM, with promising potential for practical applications like solar-energy storage, sensors, smart switches, and soft actuators, especially at the microscopic level. This molecule exhibits E-Z photoisomerization characteristics in both solution and solid states, with a conversion rate of 63.9% in the solution state and 49.5% in the solid state. Under irradiation with 365 nm light, (E)-PADM powder crystals undergo a significant photoinduced melting phase transition, transforming from a highly ordered crystalline state to an amorphous non-crystalline state. We have obtained nanowire crystals using solvent annealing to optimize this photoinduced melting process. The rate of photoinduced melting has been enhanced by more than 9000 times due to the reduction in crystal size. Our results highlight the critical impact of scale effects on photoinduced melting responses and provide new ideas for designing and optimizing photoresponsive materials at the microscopic level, inspiring further research and development in this field.”
5) The novelty and practical applications of the presented data must be highlighted.
Response: We are grateful for the reviewer's suggestions. We highlight the novelty and practical applications in the both introduction and conclusion parts, which are as follows:
“Our results demonstrate that the size reduction dramatically accelerates the speed of photoinduced phase transitions in crystals. Our results offer a straightforward approach to achieving dramatic photoresponsive behaviors in organic molecular crystals at the microscopic level, in contrast to those at the macroscopic scale. The rapid phase transition feature in nanoscopic crystals can potentially be applied in future miniature intelligent switches and energy transducers.”
“We have successfully designed and synthesized a new photoactive phenyl-based molecule, (E)-PADM, with promising potential for practical applications like solar-energy storage, sensors, smart switches, and soft actuators, especially at the microscopic level.”
“Our results highlight the critical impact of scale effects on photoinduced melting responses and provide new ideas for designing and optimizing photoresponsive materials at the microscopic level, inspiring further research and development in this field.”

Reviewer 3 Report
Comments and Suggestions for Authors
Authors of this study have claimed that they studied a new photoactive molecular crystal made from (E)-2-(3-phenyl-allylidene)malonate, which when exposed to ultraviolet (UV) light at 365nm resulted an E-to-Z photoisomerization in liquid solution and a crystal-to-liquid phase transition in solid crystals. It was shown that a dramatic collapse of molecular crystals into droplets can be made possible via photochemical reactions and phase transitions. However, the quality of English should be highly improved since there are grammatical errors starting from the second line of the abstract. In addition, the following issues should be addressed carefully:
1. The abstract of the paper is very much synonymous with the last paragraph of the introduction, which should not be. The latter should deal with the core objective of the study, but surely not observations of any kind.
2. Quality of most pictures are no good. High quality images should be provided. Moreover, the text in the captions should be transparent. For example, it is not clear what this distance 7.063 angstrom in Fig. 3(a) refer to?
3. The study lacks computational verification of observed chemical shifts and UV-vis spectral features (viz. Fig. 1), which should not be.
4. What is a Miller crystal plane/face? To my knowledge, any crystal plane should be described by a Miller index. Accordingly, the text in the ms should be properly written.
5. What are the melting points of the crystals examined?
Comments on the Quality of English Language--
Author Response
Reviewer 3:
Authors of this study have claimed that they studied a new photoactive molecular crystal made from (E)-2-(3-phenyl-allylidene)malonate, which when exposed to ultraviolet (UV) light at 365nm resulted an E-to-Z photoisomerization in liquid solution and a crystal-to-liquid phase transition in solid crystals. It was shown that a dramatic collapse of molecular crystals into droplets can be made possible via photochemical reactions and phase transitions. However, the quality of English should be highly improved since there are grammatical errors starting from the second line of the abstract. In addition, the following issues should be addressed carefully:
i) The abstract of the paper is very much synonymous with the last paragraph of the introduction, which should not be. The latter should deal with the core objective of the study, but surely not observations of any kind.
Response: we sincerely thank for the reviewer for the detailed suggestion. Therefore, we have updated the last paragraph of introduction, which is follow:
“Herein, we designed and synthesized a novel butadiene derivative molecule named (E)-2-(3-phenyl-allylidene)malonate ((E)-PADM). When exposed to ultraviolet (UV) light (365 nm), bulk polycrystals of (E)-PADM with dimensions larger than 1 millimeter in length and width undergo a slow crystal-to-liquid phase transition, partially converting into liquid after approximately 2.5 hours of light exposure. The phase transition rate significantly increases when the bulk crystals are ground into smaller powder crystals, producing more liquid in just a few minutes. However, under the same light exposure conditions, nanoscopic (E)-PADM wires completely melt into liquid within one second, approximately 9000 times faster than their bulk counterparts. Our results demonstrate that the size reduction dramatically accelerates the speed of photoinduced phase transitions in crystals. Our results offer a straightforward approach to achieving dramatic photoresponsive behaviors in organic molecular crystals at the microscopic level, in contrast to those at the macroscopic scale. The rapid phase transition feature in nanoscopic crystals can potentially be applied in future miniature intelligent switches and energy transducers.”
ii) Quality of most pictures are no good. High quality images should be provided. Moreover, the text in the captions should be transparent. For example, it is not clear what this distance 7.063 angstrom in Fig. 3(a) refer to?
Response: We agree with the reviewer that the quality of the pictures should be improved. Thus, we have updated the pictures. The distance 7.063 Å in Fig. 3a refers to the distance between the nearest double carbon-carbon bond, which has been mentioned in the context. To avoid ambiguity, we removed the relevant data from Figure 3a.
iii) The study lacks computational verification of observed chemical shifts and UV-vis spectral features (viz. Fig. 1), which should not be.
Response: We are grateful for the suggestion. We have added computational verification in Figure S10, which contains the calculated UV-Vis absorption spectra of both E and Z-isomers and the predicated 1H NMR spectra of E and Z-isomers.
iv) What is a Miller crystal plane/face? To my knowledge, any crystal plane should be described by a Miller index. Accordingly, the text in the ms should be properly written.
Response: We are very grateful for the reviewer's suggestions. We have modified the relevant content in the manuscript. We changed the Miller crystal plane into the Miller index. We used a crystal facet instead of a Miller crystal plane for some specific cases.
v) What are the melting points of the crystals examined?
Response: The crystal melting point mentioned in the manuscript was obtained from the melting peak in the DSC curve, which is around 68℃ for the (E)-PADM. We also verified it using a melting point apparatus, and the results were consistent with the DSC data.

Round 2
Reviewer 2 Report
Comments and Suggestions for Authors
The revised manuscript entitled “Size Reduction to Enhance Crystal-to-Liquid Phase Transition Induced by E-to-Z Photoisomerization Based on Molecular Crystals of Phenylbutadiene Ester”, resubmitted by Fei Tong and co-workers for reconsideration for publication in the MDPI journal Materials presents higher level than the first submission. The authors have made the required corrections and added additional explanations. Therefore, I consider the revised manuscript suitable for publication in the MDPI journal Materials.
Reviewer 3 Report
Comments and Suggestions for Authors
Authors have revised their paper based on my comments, and replied to all of them. I believe that work may be interesting to the readers of the journal.
Comments on the Quality of English Language--